# An Assessment of Existing Sport-Related Concussion Guidelines in Ireland: The Need for a United Approach

**DOI:** 10.3390/jfmk9020101

**Published:** 2024-06-10

**Authors:** Ayrton Walshe, Ed Daly, Lisa Ryan

**Affiliations:** Department of Sports, Exercise, and Nutrition, Atlantic Technological University, H91 T8NW Galway City, Ireland; ayrton.walshe@research.atu.ie (A.W.); ed.daly@atu.ie (E.D.)

**Keywords:** Irish, sport, concussion, consensus, protocols

## Abstract

In 2014, an Irish parliament white paper called for greater addressing of sport-related concussions (SRCs) in Ireland, requesting the adoption of the Concussion in Sport Group’s (CISGs) guidelines and greater consistency in SRC return to play (RTP) management. Ten years later, it is unclear how these requests have been addressed. Recently, the United Kingdom’s government centralised guidelines to one SRC document for all grassroots sports. This study aimed to investigate all publicly available SRC guidance in Irish sports and national governing bodies (NGBs) to determine if centralised guidelines are warranted. Sport Ireland and the Irish Federation of Sports were searched for all recognised NGBs and sports in Ireland. Websites were searched for any information pertaining to SRCs and data were extracted and collated in Microsoft Excel. In total, 15 of 83 sports and/or NGBs included SRC guidance, nine of which provided RTP protocols. Various iterations of the CISGs guidance and tools were implemented. Several sports with a documented SRC risk had no guidelines present. The findings indicate disjointed and outdated guidance across Irish sport. Additionally, there are sports with a documented concussion risk that have no SRC guidance available. This study provides support for centralised guidelines to be adopted in Irish grassroots sports.

## 1. Introduction

A sport-related concussion (SRC) is a mild traumatic brain injury (mTBI) that can be defined as a disruption of brain function and metabolic cascade. This disruption can be caused by direct force to the head, whereby forces are transmitted upward to the head resulting in transient symptoms in athletes [1]. This injury may occur as two players collide in tackle situations, when an athlete attempts to complete a header, or simply as an athlete falls during sport participation. As such, opportunities for SRCs to occur are frequent and many sports are likely to have a risk of SRCs. Signs and symptoms of an SRC may not always be transient and can extend into long-term symptomology known as post-concussive syndrome with varying rates reported in the literature [2,3]. An isolated SRC may not affect the brain at a structural level and therefore cannot be seen using standard neuroimaging used to investigate brain health and function [4]. However, there is emerging evidence to highlight the long-term consequences of repeated head impacts which may be linked to the length of playing careers and cumulative impact load in athletes [5,6].

Currently, there has been limited injury surveillance research of SRCs in Irish sports. For example, ladies Gaelic football and camogie have 200,000 and 100,000 active participants yet both sports have reported 11 and two diagnosed SRCs across 10 years, respectively, in the current literature [7]. The Irish Rugby Football Union’s (IRFUs) Injury Rugby Injury Surveillance (IRIS) project is the most prominent injury surveillance survey in place in Ireland; however, only seventeen male and four female teams participated in its 2023 report [8].

As knowledge growth and research continues, stakeholders in sports are beginning to implement SRC guidelines to protect both athletes and their organisations. In the Republic of Ireland, these initiatives were primarily driven by an Irish parliament white paper calling on national governing bodies (NGBs) to better address concussions. The paper specifically requested NGBs to adopt the Concussion in Sport Group’s (CISGs) guidelines, and create greater consistency in return-to-play (RTP) guidance (Figure 1) [9]. The CISG is now responsible for informing many of the guidelines that are adopted by Irish NGBs but the formulation of such guidelines can vary across sport, level of participation, and the country in question [1].

The CISG proposes standardised multi-modal assessment tools for SRCs known as the Sport Concussion Assessment Tool (SCAT), this assessment can only be conducted by medical professionals. An alternative concussion recognition tool (CRT) is available for non-medical professionals [10,11]. The SCAT is preferred over singular assessment tools [12], and both tools are warranted as SRCs can often be non-disclosed by athletes [13,14]. At present, it is unclear how many NGBs currently adopt, recommend, or distribute these tools within their sports. The United Kingdom government recently took further steps and published centralized SRC guidelines for all grassroots sports to ensure adequate athlete welfare and to prevent substandard guidelines from being implemented by NGBs [15].

This is not the first time a government has taken action to protect athletes. For example, within the United States of America (USA) rulings, such as the Lystedt laws, have enforced greater athlete welfare and SRC reporting in youth athletes in the state of Washington, which mobilised nationwide reforms in SRC management [16].

A previous narrative review was published on Irish SRC guidelines in 2016 [17] but the described methodology was a conference proceeding and thus was limited in detail and scope. It is also unclear if such guidelines have been updated since this study was published, and it is unclear which sports do not have available SRC guidelines. Therefore, the current study aims to evaluate and describe all available SRC guidance for Irish sporting NGBs to determine if the guidance is in line with current best practices and to ascertain if centralised government guidelines are warranted in Irish sports.

## 2. Materials and Methods

Ethical approval was not required for this study. Sport Ireland’s (https://www.sportireland.ie/national-governing-bodies/ngb-contact-finder, Accessed: 12 November, 2023) and the Federation of Irish Sport’s (https://www.irishsport.ie/ngbs/, Accessed: 12 November, 2023) websites were searched to find all currently recognised sports and associated NGBs in Ireland. Each of these sports and/or NGBs’ websites was individually searched, and searches were made on Google Chrome with the NGB’s title and ‘concussion protocol’ or ‘concussion guidelines’ to prevent false negatives in reporting. NGBs were not directly contacted as the purpose was to quantify and describe the current publicly available data only.

This assessment of current guidelines was conducted to determine if there is a need for national guidelines to support all recognised sporting organisations. Thus, to focus on sports with the highest participation, or documented SRC rates were not justified. Additionally, in many cases, the incidence of SRCs in Irish sports is currently unknown.

Data were then extracted from each of these websites pertaining to; (1) the presence of any SRC guidance, (2) the date of most recent guidance, (3) the source of guidance, (4) red flags, (5) clinical tools, (6) graduated RTP, and (7) marketing materials. Data are synthesised and presented below.

## 3. Results

### 3.1. Summary Analysis

In total, 83 sports and/or NGBs were identified with 15 detailing information regarding SRCs. These included collision, contact, and non-contact sports with guidelines dated from 2015–2022 and large variability in the presence of landing pages or specific links to SRC information (*n*= 9), marketing resources (*n* = 4), videos (*n* = 4), and additional external links (*n* = 5) existed. A summary of all findings are available in Appendix A, and the included sports is in Table A1. An expanded discussion of relevant sports with no SRC guidelines online (*n* = 68) will be included in the discussion but are listed in Table A2.

A summary of clinical assessments and SRC recognition tools currently available is provided in Table A3. Of the included sports, information was primarily sourced from the CISG Concussion Consensus Statements. As such, tools recommended for SRC recognition were the original CRT1 (*n* = 3), CRT5 (*n* = 5), a modified CRT (*n* = 1), CRT6 (*n* = 2), or no tool identified (*n* = 5). Regarding clinical assessment tools for SRCs, the SCAT6 (*n* = 2), SCAT5/Child SCAT5 (*n* = 4), SCAT3 (*n* = 4), Pocket SCAT2 (*n* = 1), modified and unmodified Maddocks questions (*n* = 2), and a mobile application which was unavailable at the time of data collection (*n* = 1) were identified. Two NGBs did not provide a recommended tool for SRC assessment (*n* = 2). The Gaelic Athletic Association (GAA) [18] and IRFU were the only bodies to include the CRT6, SCAT6, and Child SCAT6 on their landing page, however, the GAA still had the CRT5 and SCAT5 recommended within their guidelines. The SCAT6/Child SCAT6 and CRT6 are the most up-to-date CISG tools based on the 6th concussion consensus [1]. Basketball Ireland maintains an SRC landing page with minimal information, instead providing a link to a mobile application for screening and guidance through SRC-RTP [19]. This application could not be accessed or downloaded during data collection and thus at the time of writing no publicly available guidance is provided for this sport.

### 3.2. SRC Red Flags

Detailed SRC guidelines should include specific statements regarding the red flags for more severe forms of TBI or spinal injury. However, only the IRFU’s [20] and GAA’s [18] guidelines, which were also adopted by the Ladies Gaelic Football Association [21] (LGFA) and Camogie Association [22], discuss red flags. Where available, such red flags are included within CRT and SCAT assessment tools. A summary of the aforementioned red flags is available in Table A4.

### 3.3. Graduated RTP across Sports

Graduated RTP protocols offer a structured, stepwise pathway to recovery following an SRC. Of the 15 sports or NGBs included in this analysis, nine provided guidance on RTP following an SRC (Table A5). The protocols provided broadly follow the six-stage GRTP pathway presented by the CISG group with some individual variation; (1) Symptom-limited activity, (2) Aerobic exercise, (3) Individual sport-specific exercise, (4) Non-contact training drills, (5) Full-contact practice, and (6) Return to sport. Each of these sports advised medical clearance prior to returning to full participation in their sport. Only the GAA [18] provided a specific GRTP protocol for female athletes (as adopted by the LGFA [21] and Camogie Association [22]), and only the Football Association of Ireland (FAI) did not provide specific GRTP guidance for youths or a minimum stand-down period following an SRC [23].

American Football Ireland [24] has adopted a 2020 version of the IRFU’s GRTP protocol, which was informed by the 5th concussion consensus. These protocols require a 21-day stand-down period for adult players, the longest minimum stand-down period for adult athletes in Irish sports. This extends to 23 days for those aged 6–20 to allow an additional day for both increased aerobic exercise (Stage 2) and rugby-specific exercise (Stage 3). Their protocol also divides stage one into two parts (1A = Symptom-limited activity; 1B = Symptom-limited exercise) and requires athletes to be asymptomatic before entering stage two. The 2024 edition of the IRFU’s protocol has removed the above step [20], outlining the CISG’s clear six-stage pathway, and as mentioned above now recommends medical clearance at stage 4 (non-contact), a stage earlier than their previous GRTP protocol (stage 5).

The GAA GRTP protocols [18] (also adopted by the LGFA [21] and Camogie Association [22]) provide the only sex-specific protocol currently available in Ireland. The protocols are derived from the 5th concussion consensus statement. For male athletes, GRTP is implemented across six stages: (1) No activity for 24–48 h, (2) Light activity, (3) Sport-specific exercise, (4) No contact training drills, (5) Full contact practice, and (6) RTP. Steps 2–5 require 24 h between stages and a minimum of seven days stand down is required before returning to play. For female and youth athletes, stage one requires 48 h minimum and at least four days between stages 2–4 resulting in a 15-day minimum stand-down period. For all athletes, if there is an increase or development of symptoms during RTP, athletes are advised to drop back to the previous asymptomatic level and try to progress following a further 24 h of rest.

The Irish Flying Disc Association also adheres to the CISG protocol [25], with a minimum of 24 h between stages and a suggested seven days before returning to play for adult athletes. For youth players, a minimum of 14 days rest is suggested for stage one before entering stage 2–6. If there is an increase or development of symptoms during RTP, athletes are advised to drop back to the previous asymptomatic level and try to progress following a further 24 h of rest. Within the Irish Amateur Wrestling Association [26] and Hockey Ireland [27], a five-stage RTP is implemented as there is a direct transition from non-contact to full RTP. Adult athletes begin with 24 h rest followed by stages 2–5 with 24 h minimum between stages. A lack of clarity exists here as this would equate to a six days post-injury but later, a seven-day RTP for adults is stated. For youth athletes, a minimum rest of two weeks is required before completing stages 2–5 with 48 h between stages and equates to 23 days post-injury RTP.

The FAI protocol adopts their RTP protocol from the CISG but does not state a minimum standdown time or provide guidelines for youth or female athletes [23]. Their protocols also give no guidance as to what to do if symptoms progress or develop during stages of the GRTP.

## 4. Discussion

The aim of this study was to investigate the presence, variability, and quality of SRC guidelines across Irish sports and NGBs. Guidelines were available for 15 sports (Table A1), with the IRFU [20] and GAA [18] providing the most robust guidelines in Ireland. At present, only the GAA [18] and IRFU [20] have included information (Table A3) from the 6th concussion consensus statement (the SCAT6 and CRT6 tools). As such, some key statements and recommendations are now outdated in many of the NGBs guidance documents. There were also notable omissions from the analysis as sports with documented SRC risks had no available information provided. These are discussed in further detail below.

### 4.1. Findings in Context of 6th Consensus Statement and Current Literature

The 6th concussion consensus statement provides the most recent foundation for SRC guidance in athletic populations [1], and much of the guidance in Irish sport is based on previous iterations of these statements. Although the COVID-19 pandemic did delay the 6th consensus which took place in October 2022, there is little justification for NGBs to not update their guidelines and assessment tools in line with this guidance. To date, the IRFU is the only NGB to fully update its guidelines to reflect the 6th concussion consensus; however, this was only completed at the time of final draft preparation in early 2024 [20].

Given the dynamic nature of knowledge and the expansion of research in this area some of the current guidance provided is unwarranted. In particular, previous guidance of absolute rest until symptom resolution beyond 24–48 h post-SRC [28] is now not advised as this can lead to suboptimal recovery [1]. Outdated RTP protocols advised regression of SRC-RTP to a previous stage in the presence of symptoms [28] however present guidance now recommends continuation provided symptom exacerbation is mild and brief [1]. A key change in viewpoint by CISG and guidelines occurred between statements regarding mean RTP timelines, this was previously stated to be within 7–10 days (Table A5); however, the 6th consensus statement recognises an unrestricted mean RTP of 19.80 days (95% CI = 18.80–20.70 days) [29]. These points provide a small snapshot of key differences between current knowledge and guidelines, highlighting the need for better guidance and protection for Irish amateur athletes and regular examination of best practices.

### 4.2. Epidemiology in Sports with No SRC Guidance

The absence of publicly available SRC guidelines for NGBs with documented SRC risks is concerning, see Table A2. In youth rugby league, there is less available literature than in rugby union but rates of 4.60 and 14.70 SRCs per 1000 match hours have been identified [30]. In Super League, Championship, and Academy Rugby League rates of 15.50, 10.50, and 14.30 per 1000 player-match hours were identified between 2016–2022 [31]. For female athletes, rugby league has been observed to have higher SRC rates (10.30 per 1000 match hours) than both rugby union (2.80 per 1000 match hours), and rugby 7 s (8.90 per 1000 match hours) [32].

With respect to combat sports, in mixed-martial arts (MMA) SRC rates of 15.70% and 8.60% have been observed for amateurs and professionals, respectively, in the United States (*p* < 0.05) [33]. The exclusion of MMA is discussed in the limitation section below. In boxing, pooled rates of 12.30% (95% CI = 8.70–15.90%) were identified. Individually, higher rates were identified in professional boxing compared to amateur boxing but no significant difference existed in pooled analysis (*p* > 0.05) [34]. Pooled data from high school judo athletes found risk rates of 11.55% (2035 athlete seasons) and 7.07% (3097 athlete seasons) for female and male athletes, respectively. SRCs are clearly occurring in adult Judo as self-reported by athletes [35] but the precise rates of these are currently unclear. This rate is unlikely to be low given the documented evidence of catastrophic head injuries within the sport [36]. At the 2018 World Taekwondo Junior championships, only three diagnosed SRCs were recorded resulting in rates of 2.06 and 1.31 per 1000 AEs for male and female athletes, respectively [37]. Historically, rates have varied substantially in adults in the published literature which in part may be linked to the variety of SRC definitions used in such studies [38]. However, a 2017 systematic review found rates of 13.8/1000 athletic exposures and 12.1/1000 athletic exposures for women, respectively [39].

Cycling has an inherent risk of collision and fall-based accidents which predispose athletes and the general population to SRC risk. Determining true SRC rates in cycling sports is complicated given the variability in terrain, conditions, and participation numbers in competition. Rates of 9.10% and 7.80% were found in Belgian race competitions in 2002 and 2012, respectively [40], while self-reported questionnaires revealed that 23.8% of cyclists (*n* = 999) had previously sustained an SRC [41]. Similar to cycling, driving, and motorcycling-related SRCs affect both athletes and the general population. Deakin et al. have summarised the available literature from multiple two and four-wheeled competitions including unpublished data from the British Touring Car Championships which shows original SRC rates of 0–1 SRCs each season in the early 1990s but now stands at 6–11 SRCs per year in recent seasons [42]. Although this increase must be considered in the context of improved assessment criteria, education, and awareness of SRCs, these findings still justify the need for guidance on best practices in these sports.

### 4.3. Toward Centralisation and Practical Applications

The findings of the present study indicate that many Irish NGBs may not place SRCs as a priority or might not be aware of best practices. It is logical to assume some sports may have a negligible risk of SRCs, but many others do not. Implementation of basic protocols and guidance in line with CISG requires minimal time or financial resources and thus should not be difficult to address. More appropriately, an Irish replication of the UK’s government-driven protocols for grassroots sport should be considered to prevent bias or poor attitudes in the formulation of such protocols. Many of the sports included placed emphasis on anti-doping, despite recorded rates being minimal or unknown [43]. If such emphasis can be placed on a contemporary issue like anti-doping, there is little reason for SRCs not to be also addressed within Irish sports.

This study provides a justification for the centralisation of grassroots SRC guidelines in Ireland, due to the variability and sub-standard guidance provided by some sports, and the absence of any SRC guidance by others. There is clear evidence of the consequences of SRCs and repeated head impacts in athletic populations [5,6], and all efforts must be made to ensure athletes do not RTP until a full recovery has been made. The most reliable way to achieve this is by enforcing minimum standdown periods for athletes as many athletes have been shown to non-disclose and attempt to bypass RTP guidelines [13].

Athletes may also compete in multiple sports at the grassroots level, particularly youth or adolescent athletes [44]. Therefore, conflicting guidance must be navigated regarding SRC-RTP. For example, if an adult male athlete competing in rugby union and soccer sustains an SRC, they can return to soccer within one week but must wait a further two weeks before returning to rugby union. SRCs do not discriminate between sports, this may lead to conflicting guidance and adds additional complexity to SRC-RTP. If centralised guidelines could be established via government incentives, it would create a more cohesive environment to disseminate one key message by many NGBs. This would be prudent and pertinent as there are many knowledge gaps in RTP guidelines amongst Irish coaches, allied healthcare practitioners, and general practitioners [13]. Such an approach may allow for resources to focus on the dissemination and upskilling of medical and non-medical practitioners in the implementation of a unified SRC-RTP protocol.

### 4.4. Limitations

Limitations are present in this study. As previously mentioned, The IRFU currently had updated guidance prepared during manuscript preparation and other NGBs may also be in the process of doing so. Despite this, given the variability of findings in the context of the 5th consensus statement which was published in 2017, it is likely this is not the case for many NGBs. To exemplify this point, the FAI updated its concussion guideline page into a downloadable file in April 2024 [23] but did not include any new information from the 6th concussion consensus. The inclusion of sports within this study was dependent on their registration with Sport Ireland or the Federation of Irish Sport, therefore many sports such as MMA or Brazilian Jiu Jitsu have been excluded. It is unclear how and if these and other excluded sports would implement SRC guidelines. It is also important to reiterate that some sports may have SRC guidelines, but do not make them publicly available.

This study has placed emphasis on the CISG given their references in the original Irish parliament paper and role in current Irish SRC guidance. However, the body and members have faced criticism in recent years for multiple reasons issues such as plagiarism [45] and conflicts of interest [46]. The CISG has attempted to provide greater transparency during the process of their 6th consensus statement and specific criticised members were not involved in this process [1] The CISGs statements provide free, research-based evidence and a high-quality assessment tool for SRCs and this work is greatly appreciated at all levels of sport. Therefore, this guidance is recommended until an independent process (such as the UK’s) can be lobbied for, conducted, and implemented [15].

## 5. Conclusions

There is substantial variability in the quality of current SRC guidance provided by Irish sporting NGBs, much of this guidance has yet to be updated and informed by current literature and/or the 6th concussion consensus statement by CISG. Many sports with a documented SRC risk have no available SRC guidance which should be addressed immediately. The current findings do indicate that centralised guidelines may be warranted in Irish grassroots sports to better protect athlete welfare.

## Figures and Tables

**Figure 1 jfmk-09-00101-f001:**
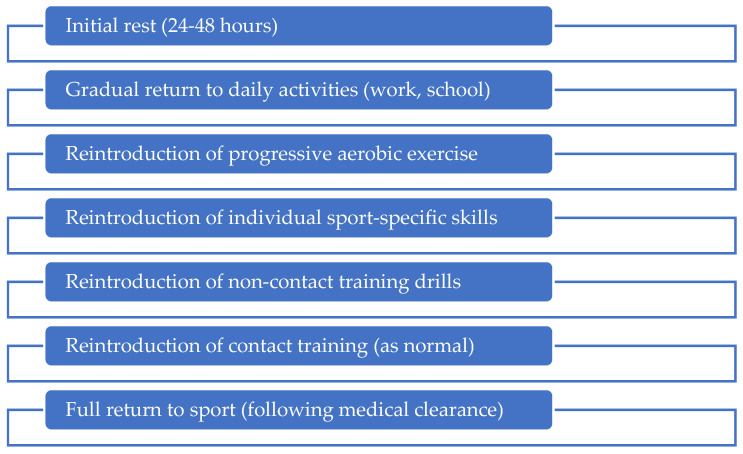
Summary of the sport-related concussion return to play guidance as per the Concussion in Sport Group [1].

## Data Availability

All data are available within this manuscript and associated references.

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
