# Peer review of "An Assessment of Existing Sport-Related Concussion Guidelines in Ireland: The Need for a United Approach"

_jfmk, 2024, doi:10.3390/jfmk9020101_

Round 1
Reviewer 1 Report
Comments and Suggestions for Authors
An assessment of existing sport-related concussion guidelines in Ireland; the need for a united approach, submitted to J. Funct. Morphol. Kinesiol. It is a good work with interesting insights in the context of head impacts in athletic populations. I appreciate the concern of authors with relation the protection for Irish athletes. The study is worthy of publication.
· The introduction provides a relevant background to the study; However, Please, the authors should provide some neuro mechanism underlying traumatic brain injury. This needs to be clearer to the reader (to emphasize the necessity of attention to the problem of concussion).
· Experimental design, methods and results seems proper, however, I suggest the inclusion of more figures. For example: what clinical assessments/recognition tool is more frequent among organizations. Results of Table A3 could be illustrated.
· Regarding “the need for a united approach”. I would like to know the authors' line of reasoning about this.
· Didactics would improve with the inclusion of a figure (some scheme drawn by the authors) tying up the chronological events of this area (timeline of creation of laws, organizations, guidelines).
Author Response
Reviewer 1:
Comment:: An assessment of existing sport-related concussion guidelines in Ireland; the need for a united approach, submitted to J. Funct. Morphol. Kinesiol. It is a good work with interesting insights in the context of head impacts in athletic populations. I appreciate the concern of authors with relation the protection for Irish athletes. The study is worthy of publication.
Response: Thank you for taking time to review our work and we really appreciate your feedback and suggestions.
Comment: The introduction provides a relevant background to the study; However, Please, the authors should provide some neuro mechanism underlying traumatic brain injury. This needs to be clearer to the reader (to emphasize the necessity of attention to the problem of concussion).
Response: To better describe the mechanism of how this brain injury occurs, the following has been included in the introduction (Line 29-32: This injury may occur as two players collide in tackle situations, when an athlete attempts to complete a header, or simply as an athlete falls during sport participation. As such, opportunities for SRCs to occur are frequent and many sports are likely to have a risk of SRCs.). Regarding neurological mechanisms of brain injury, these have been discussed in the initial definition (Line 26-29) and the long-term consequences briefly mentioned (Line 36-39). This study aims to focus on the rehabilitation of an isolated SRC, which is not of concern once managed appropriately and thus we do not want to further discuss the problems of the brain injury.
Comment: Experimental design, methods and results seems proper, however, I suggest the inclusion of more figures. For example: what clinical assessments/recognition tool is more frequent among organizations. Results of Table A3 could be illustrated.
Response: Thank you for this comment. We did try illustrate A3 but we feel it limits the clarity of findings and disassociates the tools from the specific NGBs discussed. The exact tools used by each organisation is just as important for us as the number of each tool used. Based on your feedback we did however included two figures (Figure 1-2) within the introduction to help readers better understand from the start what best practice tools might include.
Comment: Regarding “the need for a united approach”. I would like to know the authors' line of reasoning about this.
Response: As discussed (Line 66-69), the UK recently created a united protocol for all grassroots sport to protect athletes (Example now also included in Figure 2). We believe our findings highlight this should be replicated in Ireland given the variability in presence and quality of protocols at this time (Subsection 4.3).
Comment: Didactics would improve with the inclusion of a figure (some scheme drawn by the authors) tying up the chronological events of this area (timeline of creation of laws, organizations, guidelines).
Response: Although we appreciate this idea we have included two figures within the introduction based on your feedback and another within the introduction would be too much at this time.
Reviewer 2 Report
Comments and Suggestions for Authors
Congratulations to the authors for their work. The topic seems relevant to me because of the high health risk posed by concussion trauma. The study is limited to the Republic of Ireland but gives valuable information about current sport institutions that have or do not have concussion prevention policies. From my point of view, the introduction perfectly orients the problem to be addressed and clearly defines the objective of the study. It is also in line with recent studies of the same research group (13) in which the same problem is approached from a complementary point of view to the current one. The methodology employed seems to me to be adequate for the proposed objectives and the results are well identified in the attached tables. The discussion separates different clearly defined sections and relevant practical contributions are made. Limitations of the study are also written although these are rather temporary. I really doubt that my contributions can improve the study as it is very well elaborated and better written.
We hope that the observations made will reach the sports institutions that still have room for improvement in order to avoid concussion in sports.
Best look.
Author Response
Comment: Congratulations to the authors for their work. The topic seems relevant to me because of the high health risk posed by concussion trauma. The study is limited to the Republic of Ireland but gives valuable information about current sport institutions that have or do not have concussion prevention policies. From my point of view, the introduction perfectly orients the problem to be addressed and clearly defines the objective of the study. It is also in line with recent studies of the same research group (13) in which the same problem is approached from a complementary point of view to the current one. The methodology employed seems to me to be adequate for the proposed objectives and the results are well identified in the attached tables. The discussion separates different clearly defined sections and relevant practical contributions are made. Limitations of the study are also written although these are rather temporary. I really doubt that my contributions can improve the study as it is very well elaborated and better written.
We hope that the observations made will reach the sports institutions that still have room for improvement in order to avoid concussion in sports.
Best look.
Response: Thank you very much for your comments and for taking time to review our study. All the best.
Reviewer 3 Report
Comments and Suggestions for Authors
Dear authors,
You take up an important topic, a pre -led analysis can be effective used in practice and pre -pre -charged to make you can do recovery and welds, athletes. Research requires how to look in the future in a guide of profile actions that can have a risk of risk (SRC) and MTBI
The work can be published after minor corrections -you can release the tables to make them more readable
Author Response
Dear authors,
You take up an important topic, a pre -led analysis can be effective used in practice and pre -pre -charged to make you can do recovery and welds, athletes. Research requires how to look in the future in a guide of profile actions that can have a risk of risk (SRC) and MTBI
The work can be published after minor corrections -you can release the tables to make them more readable
Response: Thank you for your feedback and we agree this work can drive further research in this area. As we understand, the editors will lead the formatting of tables and will instruct on the final formatting (previously with JMFK this was JPEG conversion) after acceptance. We hope this will help make the tables more readable but will await their instruction on this.
Reviewer 4 Report
Comments and Suggestions for Authors
I reviewed the article by Walshe and colleagues titled “An assessment of existing sport-related concussion guidelines in Ireland; the need for a united approach”. The manuscript aims to investigate all publicly available sport-related concussion guidance in Irish sports and national governing bodies to determine if centralized guidelines are warranted. The researchers conducted the study well, utilized appropriate methods, and presented clear results.
I have a question and suggestion to be included in the text. Are there any recommendations when teaching soccer to children? This is because it is inherent to this sport to use the head voluntarily to touch the ball. Therefore, this is taught since childhood. However, as children do not know how to execute correctly, this causes a greater impact on the head. Consequently, it is essential to have recommendations for teaching this foundation, as children still have their organic structures under development, which could lead to future complications.
Author Response
Comment: I reviewed the article by Walshe and colleagues titled “An assessment of existing sport-related concussion guidelines in Ireland; the need for a united approach”. The manuscript aims to investigate all publicly available sport-related concussion guidance in Irish sports and national governing bodies to determine if centralized guidelines are warranted. The researchers conducted the study well, utilized appropriate methods, and presented clear results.
I have a question and suggestion to be included in the text. Are there any recommendations when teaching soccer to children? This is because it is inherent to this sport to use the head voluntarily to touch the ball. Therefore, this is taught since childhood. However, as children do not know how to execute correctly, this causes a greater impact on the head. Consequently, it is essential to have recommendations for teaching this foundation, as children still have their organic structures under development, which could lead to future complications.
Response: Thank you so much for taking the time to read and review our study, we appreciate your feedback and we hope the following we appreciate your question. At this time, there is a transition away from permitting children to head the ball through restrictions at specific age groups (e.g. 12 and younger) in countries such as the UK. You mention how greater impact on the head may come from an inability to head the ball correctly, but to play devils advocate: What happens when we don’t allow children to learn to head the ball, does this increase their risk due to having less experience and skill development when they start heading? Or what about more elite contexts where a young soccer athlete (Jude Bellingham for example) makes their professional senior debut at 15, but in the future such a player will only have two years of heading experience. Both our hypotheses could be true or false, but until we await research on the implications of such interventions we would not include it within the scope of this study. We hope this has helped provide an understanding of the lead authors stance on this issue.